# ADAPTIVE LEARNED BLOOM FILTER (ADA-BF): EFFICIENT UTILIZATION OF THE CLASSIFIER

## ABSTRACT

Recent work suggests improving the performance of Bloom filter by incorporating a machine learning model as a binary classifier. However, such learned Bloom filter does not take full advantage of the predicted probability scores. We proposed new algorithms that generalize the learned Bloom filter by using the complete spectrum of the scores regions. We proved our algorithms have lower False Positive Rate (FPR) and memory usage compared with the existing approaches to learned Bloom filter. We also demonstrated the improved performance of our algorithms on real-world datasets.

## 1 INTRODUCTION

Bloom filter (BF) is a widely used data structure for low-memory and high-speed approximate membership testing (Bloom, 1970). Bloom filters compress a given set $S$ into bit arrays, where we can approximately test whether a given element (or query) $x$ belongs to a set $S$, i.e., $x \in S$ or otherwise. Several applications, in particular caching in memory constrained systems, have benefited tremendously from BF (Broder et al., 2002).

Bloom filter ensures a zero false negative rate (FNR), which is a critical requirement for many applications. However, BF does not have a non-zero false positive rate (FPR) (Dillinger and Manolios, 2004) due to hashing collisions, which measures the performance of BF. There is a known theoretical limit to this reduction. To achieve a FPR of $\epsilon$, BF costs at least $n \log_2(1/\epsilon) \log_2 e$ bits ($n = |S|$), which is $\log_2 e \approx 44\%$ off from the theoretical lower bound (Carter et al., 1978). Mitzenmacher (2002) proposed Compressed Bloom filter to address the suboptimal space usage of BF, where the space usage can reach the theoretical lower bound in the optimal case.

To achieve a more significant reduction of FPR, researchers have generalized BF and incorporated information beyond the query itself to break through the theoretical lower bound of space usage. Bruck et al. (2006) has made use of the query frequency and varied the number of hash functions based on the query frequency to reduce the overall FPR. Recent work (Kraska et al., 2018; Mitzenmacher, 2018) has proposed to improve the performance of standard Bloom filter by incorporating a machine learning model. This approach paves a new hope of reducing false positive rates beyond the theoretical limit, by using context-specific information in the form of a machine learning model (Hsu et al., 2019). Rae et al. (2019) further proposed Neural Bloom Filter that learns to write to memory using a distributed write scheme and achieves compression gains over the classical Bloom filter.

The key idea behind Kraska et al. (2018) is to use the machine learning model as a pre-filter to give each query $x$ a score $s(x)$. $s(x)$ is usually positively associated with the odds that $x \in S$. The assumption is that in many practical settings, the membership of a query in the set $S$ can be figured out from observable features of $x$ and such information is captured by the classifier assigned score $s(x)$. The proposal of Kraska et al. uses this score and treats query $x$ with score $s(x)$ higher than a pre-determined threshold $\tau$ (high confidence predictions) as a direct indicator of the correct membership. Queries with scores less than $\tau$ are passed to the back-up Bloom filter.

Compared to the standard Bloom filter, learned Bloom filter (LBF) uses a machine learning model to answer keys with high score $s(x)$. Thus, the classifier reduces the number of the keys hashed into the Bloom filter. When the machine learning model has a reliable prediction performance, learned Bloom filter significantly reduce the FPR and save memory usage (Kraska et al., 2018). Mitzenmacher (2018) further provided a formal mathematical model for estimating the performance of LBF. In the same paper, the author proposed a generalization named sandwiched learned Bloom filter (sandwiched

LBF), where an initial filter is added before the learned oracle to improve the FPR if the parameters are chosen optimally.

**Wastage of Information:** For existing learned Bloom filters to have a lower FPR, the classifier score greater than the threshold $\tau$ should have a small probability of wrong answer. Also, a significant fraction of the keys should fall in this high threshold regime to ensure that the backup filter is small. However, when the score $s(x)$ is less than $\tau$, the information in the score $s(x)$ is never used. Thus, there is a clear waste of information. For instance, consider two elements $x_1$ and $x_2$ with $\tau > s(x_1) \gg s(x_2)$. In the existing solutions, $x_1$ and $x_2$ will be treated in the exact same way, even though there is enough prior to believing that $x_1$ is more likely positive compared to $x_2$.

**Strong dependency on Generalization:** It is natural to assume that prediction with high confidence implies a low FPR when the data distribution does not change. However, this assumption is too strong for many practical settings. First and foremost, the data distribution is likely to change in an online streaming environment where Bloom filters are deployed. Data streams are known to have bursty nature with drift in distribution (Kleinberg, 2003). As a result, the confidence of the classifier, and hence the threshold, is not completely reliable. Secondly, the susceptibility of machine learning oracles to adversarial examples brings new vulnerability in the system. Examples can be easily created where the classifier with any given confidence level $\tau$, is incorrectly classified. Bloom filters are commonly used in networks where such increased adversarial false positive rate can hurt the performance. An increased latency due to collisions can open new possibilities of Denial-of-Service attacks (DoS) (Feinstein et al., 2003).

**Motivation:** For a binary classifier, the density of score distribution, $f(s(x))$ shows a different trend for elements in the set and outside the set $S$. We observe that for keys, $f(s(x)|x \in S)$ shows ascending trend as $s(x)$ increases while $f(s(x)|x \notin S)$ has an opposite trend. To reduce the overall FPR, we need lower FPRs for groups with a high $f(s(x)|x \notin S)$. Hence, if we are tuning the number of hash functions differently, more hash functions are required for the corresponding groups. While for groups with a few non-keys, we allow higher FPRs. This variability is the core idea to obtaining a sweeter trade-off.

**Our Contributions:** Instead of only relying on the classifier whether score $s(x)$ is above a single specific threshold, we propose two algorithms, Ada-BF and disjoint Ada-BF, that rely on the complete spectrum of scores regions by adaptively tuning Bloom filter parameters in different score regions. 1) Ada-BF tunes the number of hash functions differently in different regions to adjust the FPR adaptively; disjoint Ada-BF allocates variable memory Bloom filters to each region. 2) Our theoretical analysis reveals a new set of trade-offs that brings lower FPR with our proposed scheme compared to existing alternatives. 3) We evaluate the performance of our algorithms on two datasets: malicious URLs and malware MD5 signatures, where our methods reduce the FPR by over 80% and save 50% of the memory usage over existing learned Bloom filters.

**Notations:** Our paper includes some notations that need to be defined here. Let $[g]$ denote the index set $\{1, 2, \cdots, g\}$. We define query $x$ as a key if $x \in S$, or a non-key if $x \notin S$. Let $n$ denote the size of keys ($n = |S|$), and $m$ denote the size of non-keys. We denote $K$ as the number of hash functions used in the Bloom filter.

## 2 Review: Bloom Filter and Learned Bloom Filter

**Bloom Filter:** Standard Bloom filter for compressing a set $S$ consists of an $R$-bits array and $K$ independent random hash function $h_1, h_2, \cdots, h_K$, taking integer values between $0$ and $R - 1$, i.e., $h_i : S \Rightarrow \{0, 1, \cdots, R - 1\}$. The bit array is initialized with all $0$. For every item $x \in S$, the bit value of $h_i(x) = 1$, for all $i \in \{0, 1, \cdots, K\}$, is set to $1$.

To check a membership of an item $x^{'}$ in the set $S$, we return true if all the bits $h_i(x^{'})$, for all $i \in \{0, 1, \cdots, K\}$, have been set to $1$. It is clear that Bloom filter has zero FNR (false negative rate). However, due to lossy hash functions, $x^{'}$ may be wrongly identified to be positive when $x^{'} \notin S$ while all the $h_i(x^{'})$s are set to $1$ due to random collisions. It can be shown that if the hash functions are independent, the expected FPR can be written as follows

$$\mathbb{E}\left(\text{FPR}\right) = \left(1 - \left(1 - \frac{1}{R}\right)^{Kn}\right)^K.$$

**Learned Bloom filter:** Learned Bloom filter adds a binary classification model to reduce the effective number of keys going to the Bloom filter. The classifier is pre-trained on some available training data to classify whether any given query $x$ belongs to $S$ or not based on its observable features. LBF sets a threshold, $\tau$, where $x$ is identified as a key if $s(x) \geq \tau$. Otherwise, $x$ will be inserted into a Bloom filter to identify its membership in a further step (Figure 1). Like standard Bloom filter, LBF also has zero FNR. And the false positives can be either caused by that false positives of the classification model ($s(x|x \notin S) \geq \tau$) or that of the Bloom filter.

It is clear than when the region $s(x) \geq \tau$ contains large number of keys, the number of keys inserted into the Bloom filter decreases which leads to favorable FPR. However, since we identify the region $s(x) \geq \tau$ as positives, higher values of $\tau$ is better. At the same time, large $\tau$ decreases the number of keys in the region $s(x) \geq \tau$, increasing the load of the Bloom filter. Thus, there is a clear trade-off.

# 3 A STRICT GENERALIZATION: ADAPTIVE LEARNED BLOOM FILTER (ADA-BF)

With the formulation of LBF in the previous section, LBF actually divides the $x$ into two groups. When $s(x) \geq \tau$, $x$ will be identified as a key directly without testing with the Bloom filter. In other words, it uses zero hash function to identify its membership. Otherwise, we will test its membership using $K$ hash functions. In other view, LBF switches from $K$ hash functions to no hash function at all, based on $s(x) \geq \tau$ or not. Continuing with this mindset, we propose adaptive learned Bloom filter, where $x$ is divided into $g$ groups based on $s(x)$, and for group $j$, we use $K_j$ hash functions to test its membership. The structure of Ada-BF is represented in Figure 1(b).

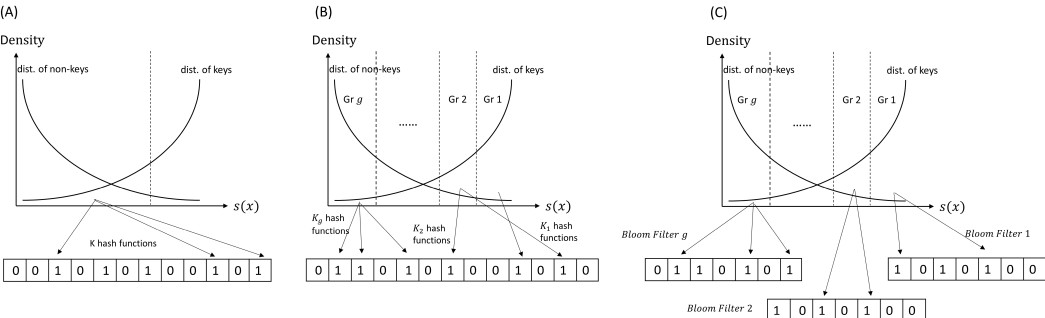

Figure 1: Panel A-C show the structure of LBF, Ada-BF and disjoint Ada-BF respectively.

More specifically, we divide the spectrum into $g$ regions, where $x \in$ Group $j$ if $s(x) \in [\tau_{j-1}, \tau_j)$, $j = 1, 2, \cdots, g$. Without loss of generality, here, we assume $0 = \tau_0 < \tau_1 < \cdots < \tau_{g-1} < \tau_g = 1$. Keys from group $j$ are inserted into Bloom filter using $K_j$ independent hash functions. Thus, we use different number of universal hash functions for keys from different groups.

For a group $j$, the expected FPR can be expressed as,

$$\mathbb{E}\left(\text{FPR}_j\right) = \left(1 - \left(1 - \frac{1}{R}\right)^{\sum_{t=1}^{g} n_t K_t}\right)^{K_j} = \alpha^{K_j} \tag{1}$$

where $n_t = \sum_{t=1}^{n} I(\tau_{t-1} \leq s(x_i|x_i \in S) < \tau_t)$ is the number of keys falling in group $t$, and $K_j$ is the number of hash functions used in group $j$. By varying $K_j$, $\mathbb{E}\left(\text{FPR}_j\right)$ can be controlled differently for each group.

Variable number of hash functions gives us enough flexibility to tune the FPR of each region. To avoid the bit array being overloaded, we only increase the $K_j$ for groups with large number of keys $n_j$, while decrease $K_j$ for groups with small $n_j$. It should be noted that $f(s(x)|x \in S)$ shows an opposite trend compared to $f(s(x)|x \notin S)$ as $s(x)$ increases (Figure 2). Thus, there is a need for variable tuning, and a spectrum of regions gives us the room to exploit these variability efficiently. Clearly, Ada-BF generalizes the LBF. When Ada-BF only divides the queries into two groups, by setting $K_1 = K$, $K_2 = 0$ and $\tau_1 = \tau$, Ada-BF reduces to the LBF.

### 3.1 SIMPLIFYING THE HYPER-PARAMETERS

To implement Ada-BF, there are some hyper-parameters to be determined, including the number of hash functions for each group $K_j$ and the score thresholds to divide groups, $\tau_j$ ($\tau_0 = 0$, $\tau_g = 1$). Altogether, we need to tune $2g - 1$ hyper-parameters. Use these hyper-parameters, for Ada-BF, the expected overall FPR can be expressed as,

$$\mathbb{E}\,(\mathrm{FPR}) = \sum_{j=1}^{g} p_j \mathbb{E}\,(\mathrm{FPR}_j) = \sum_{j=1}^{g} p_j \alpha^{K_j} \tag{2}$$

where $p_j = Pr(\tau_{j-1} \leq s(x_i | x_i \notin S) < \tau_j)$. Empirically, $p_j$ can be estimated by $\hat{p}_j = \frac{1}{m} \sum_{i=1}^{m} I(\tau_{j-1} \leq s(x_i | x_i \notin S) < \tau_j) = \frac{m_j}{m}$ ($m$ is size of non-keys in the training data and $m_j$ is size of non-keys belonging to group $j$). It is almost impossible to find the optimal hyper-parameters that minimize the $\mathbb{E}\,(\mathrm{FPR})$ in reasonable time. However, since the estimated false positive items $\sum_{j=1}^{g} m_j \alpha^{K_j} = O(\max_j(m_j \alpha^{K_j}))$, we prefer $m_j \alpha^{K_j}$ to be similar across groups when $\mathbb{E}\,(\mathrm{FPR})$ is minimized. While $\alpha^{K_j}$ decreases exponentially fast with larger $K_j$, to keep $m_j \alpha^{K_j}$ stable across different groups, we require $m_j$ to grow exponentially fast with $K_j$. Moreover, since $f(s(x)|x \notin S)$ increases as $s(x)$ becomes smaller for most cases, $K_j$ should also be larger for smaller $s(x)$. Hence, to balance the number of false positive items, as $j$ diminishes, we should increase $K_j$ linearly and let $m_j$ grow exponentially fast.

With this idea, we provide a strategy to simplify the tuning procedure. We fix $\frac{p_j}{p_{j+1}} = c$ and $K_j - K_{j+1} = 1$ for $j = 1, 2, \cdots, g - 1$. Since the true density of $s(x|x \notin S)$ is unknown. To implement the strategy, we estimate $\frac{p_j}{p_{j+1}}$ by $\widehat{\frac{p_j}{p_{j+1}}} = \frac{m_j}{m_{j+1}}$ and fix $\frac{m_j}{m_{j+1}} = c$. This strategy ensures $\hat{p}_j$ to grow exponentially fast with $K_j$. Now, we only have three hyper-parameters, $c$, $K_{min}$ and $K_{max}$ ($K_{max} = K_1$). By default, we may also set $K_{min} = K_g = 0$, equivalent to identifying all the items in group $g$ as keys.

**Lemma 1:** Assume 1) the scores of non-keys, $s(x)|x \notin S$, are independently following a distribution $f$; 2) The scores of non-keys in the training set are independently sampled from a distribution $f$. Then, the overall estimation error of $\hat{p}_j$, $\sum_j |\hat{p}_j - p_j|$, converges to 0 in probability as $m$ becomes larger. Moreover, if $m \geq \frac{2(k-1)}{\epsilon^2} \left[ \sqrt{\frac{1}{\pi}} + \sqrt{\frac{1-2/\pi}{\delta}} \right]^2$, with probability at least $1 - \delta$, we have $\sum_j |\hat{p}_j - p_j| \leq \epsilon$.

Even though in the real application, we cannot access the exact value of $p_j$, which may leads to the estimation error of the real $\mathbb{E}\,(\mathrm{FPR})$. However, Lemma 1 shows that as soon as we can collect enough non-keys to estimate the $p_j$, the estimation error is almost negligible. Especially for the large scale membership testing task, collecting enough non-keys is easy to perform.

### 3.2 ANALYSIS OF ADAPTIVE LEARNED BLOOM FILTER

Compared with the LBF, Ada-BF makes full use the of the density distribution $s(x)$ and optimizes the FPR in different regions. Next, we will show Ada-BF can reduce the optimal FPR of the LBF without increasing the memory usage.

When $p_j/p_{j+1} = c_j \geq c > 1$ and $K_j - K_{j+1} = 1$, the expected FPR follows,

$$\mathbb{E}\,(\mathrm{FPR}) = \sum_{j=1}^{g} p_j \alpha^{K_j} = \frac{\sum_{j=1}^{g} c^{g-j} \alpha^{K_j}}{\sum_{j=1}^{g} c^{g-j}} \leq \begin{cases} \dfrac{(1-c)(1-(c\alpha)^g)}{(\frac{1}{\alpha}-c)(\alpha^g-(c\alpha)^g)} \alpha^{K_{max}}, & c\alpha \neq 1 \\ \dfrac{1-c}{1-c^g} \cdot g, & c\alpha = 1 \end{cases} \tag{3}$$

where $K_{max} = K_1$. To simplify the analysis, we assume $c\alpha > 1$ in the following theorem. Given the number of groups $g$ is fixed, this assumption is without loss of generality satisfied by raising $c$ since $\alpha$ will increase as $c$ becomes larger. For comparisons, we also need $\tau$ of the LBF to be equal to $\tau_{g-1}$ of the Ada-BF. In this case, queries with scores higher than $\tau$ are identified as keys directly by the machine learning model. So, to compare the overall FPR, we only need to compare the FPR of queries with scores lower than $\tau$.

**Theorem 1:** For Ada-BF, given $\frac{p_j}{p_{j+1}} \geq c > 1$ for all $j \in [g-1]$, if there exists $\lambda > 0$ such that $c\alpha \geq 1 + \lambda$ holds, and $n_{j+1} - n_j > 0$ for all $j \in [g-1]$ ($n_j$ is the number of keys in group $j$). When g is large enough and $g \leq \lfloor 2K \rfloor$, then Ada-BF has smaller FPR than the LBF. Here $K$ is the number of hash functions of the LBF.

Theorem 1 requires the number of keys $n_j$ keeps increasing while $p_j$ decreases exponentially fast with $j$. As shown in figure 2, on real dataset, we observe from the histogram that as score increases, $f(s(x)|x \notin S)$ decreases very fast while $f(s(x)|x \in S)$ increases. So, the assumptions of Theorem 1 are more or less satisfied.

Moreover, when the number of buckets is large enough, the optimal $K$ of the LBF is large as well. Given the assumptions hold, theorem 1 implies that we can choose a larger $g$ to divide the spectrum into more groups and get better FPR. The LBF is sub-optimal as it only has two regions. Our experiments clearly show this trend. For figure 3(a), Ada-BF achieves 25% of the FPR of the LBF when the bitmap size = 200Kb, while when the budget of buckets = 500Kb, Ada-BF achieves 15% of the FPR of the LBF. For figure 3(b), Ada-BF only reduces the FPR of the LBF by 50% when the budget of buckets = 100Kb, while when the budget of buckets = 300Kb, Ada-BF reduces 70% of the FPR of the LBF. Therefore, both the analytical and experimental results indicate superior performance of Ada-BF by dividing the spectrum into more small groups. On the contrary, when $g$ is small, Ada-BF is more similar to the LBF, and their performances are less differentiable.

# 4 DISJOINT ADAPTIVE LEARNED BLOOM FILTER (DISJOINT ADA-BF)

Ada-BF divides keys into $g$ groups based on their scores and hashes the keys into the same Bloom filter using different numbers of hash functions. With the similar idea, we proposed an alternative approach, disjoint Ada-BF, which also divides the keys into $g$ groups, but hashes keys from different groups into independent Bloom filters. The structure of disjoint Ada-BF is represented in Figure 1(c). Assume we have total budget of $R$ bits for the Bloom filters and the keys are divided into $g$ groups using the same idea of that in Ada-BF. Consequently, the keys from group $j$ are inserted into $j$-th Bloom filter whose length is $R_j$ ($R = \sum_{j=1}^{g} R_j$). Then, during the look up stage, we just need to identify a query's group and check its membership in the corresponding Bloom filter.

## 4.1 SIMPLIFYING THE HYPER-PARAMETERS

Analogous to Ada-BF, disjoint Ada-BF also has a lot of hyper-parameters, including the thresholds of scores for groups division and the lengths of each Bloom filters. To determine thresholds $\tau_j$, we use similar tuning strategy discussed in the previous section of tuning the number of groups $g$ and $\frac{m_j}{m_{j+1}} = c$. To find $R_j$ that optimizes the overall FPR, again, we refer to the idea in the previous section that the expected number of false positives should be similar across groups. For a Bloom filter with $R_j$ buckets, the optimal number of hash functions $K_j$ can be approximated as $K_j = \frac{R_j}{n_j} \log(2)$, where $n_j$ is the number of keys in group $j$. And the corresponding optimal expected FPR is $\mathbb{E}(\text{FPR}_j) = \mu^{R_j/n_j}$ ($\mu \approx 0.618$). Therefore, to enforce the expected number of false items being similar across groups, $R_j$ needs to satisfy

$$m_j \cdot \mu^{\frac{R_j}{n_j}} = m_1 \cdot \mu^{\frac{R_1}{n_1}} \iff \frac{R_j}{n_j} - \frac{R_1}{n_1} = \frac{(j-1)\log(c)}{\log(\mu)}$$

Since $n_j$ is known given the thresholds $\tau_j$ and the total budget of buckets $R$ are known, thus, $R_j$ can be solved accordingly. Moreover, when the machine learning model is accurate, to save the memory usage, we may also set $R_g = 0$, which means the items in group $j$ will be identified as keys directly.

## 4.2 ANALYSIS OF DISJOINT ADAPTIVE LEARNED BLOOM FILTER

The disjoint Ada-BF uses a group of shorter Bloom filters to store the hash outputs of the keys. Though the approach to control the FPR of each group is different from the Ada-BF, where the Ada-BF varies $K$ and disjoint Ada-BF changes the buckets allocation, both methods share the same core idea to lower the overall FPR by reducing the FPR of the groups dominated by non-keys. Disjoint Ada-BF allocates more buckets for these groups to a achieve smaller FPR. In the following theorem,

we show that to achieve the same optimal expected FPR of the LBF, disjoint Ada-BF consumes less buckets. Again, for comparison we need $\tau$ of the LBF is equal to $\tau_{g-1}$ of the disjoint Ada-BF.

**Theorem 2:**   If $\frac{p_j}{p_{j+1}} = c > 1$ and $n_{j+1} - n_j > 0$ for all $j \in [g-1]$ ($n_j$ is the number of keys in group $j$), to achieve the optimal FPR of the LBF, the disjoint Ada-BF consumes less buckets compared with the LBF when $g$ is large.

## 5   EXPERIMENT

**Baselines:**   We test the performance of four different learned Bloom filters: 1) standard Bloom filter, 2) learned Bloom filter, 3) sandwiched learned Bloom filter, 4) adaptive learned Bloom filter, and 5) disjoint adaptive learned Bloom filter. We use two datasets which have different associated tasks, namely: 1) Malicious URLs Detection and 2) Virus Scan. Since all the variants of Bloom filter structures ensure zero FNR, the performance is measured by their FPRs and corresponding memory usage.

### 5.1   TASK1: MALICIOUS URLS DETECTION

We explore using Bloom filters to identify malicious URLs. We used the URLs dataset down-loaded from Kaggle, including 485,730 unique URLs. 16.47% of the URLs are malicious, and others are benign. We randomly sampled 30% URLs (145,719 URLs) to train the malicious URL classification model. 17 lexical features are extracted from URLs as the classification fea-tures, such as "host name length", "path length", "length of top level domain", etc. We used "sklearn.ensemble.RandomForestClassifier[1]" to train a random forest model. After saving the model with "pickle", the model file costs 146Kb in total. "sklearn.predict_prob" was used to give scores for queries.

We tested the optimal FPR for the four learned Bloom filter methods under the total memory budget = 200Kb to 500Kb (kilobits). Since the standard BF does not need a machine learning model, to make a fair comparison, the bitmap size of BF should also include the machine learning model size (146 Kb in this experiment). Thus, the total bitmap size of BF is 346Kb to 646Kb. To implement the LBF, we tuned $\tau$ between 0 and 1, and picked the one giving the minimal FPR. The number of hash functions was determined by $K = \text{Round}(\frac{R}{n_0} \log 2)$, where $n_0$ is the number of keys hashed into the Bloom filter conditional $\tau$. To implement the sandwiched LBF, we searched the optimal $\tau$ and calculated the corresponding initial and backup filter size by the formula in Mitzenmacher (2018). When the optimal backup filter size is larger than the total bits budget, sandwiched LBF does not need a initial filter and reduces to a standard LBF. For the Ada-BF, we used the tuning strategy described in the previous section. $K_{min}$ was set to 0 by default. Thus, we only need to tune the combination of $(K_{max}, c)$ that gives the optimal FPR. Similarly, for disjoint Ada-BF, we fixed $R_g = 0$ and searched for the optimal $(g, c)$.

**Result:**   Our trained machine learning model has a classification accuracy of 0.93. Considering the non-informative frequent class classifier (just classify as benign URL) gives accuracy of 0.84, our trained learner is not a strong classifier. However, the distribution of scores is desirable (Figure 2), where as $s(x)$ increases, the empirical density of $s(x)$ decreases for non-keys and also increases for keys. In our experiment, when the sandwiched LBF is optimized, the backup filter size always exceeds the total bitmap size. Thus, it reduces to the LBF and has the same FPR (as suggested by Figure 4(a)).

Our experiment shows that compared to the LBF and sandwiched LBF, both Ada-BF and disjoint Ada-BF achieve much lower FPRs. When filter size = 500Kb, Ada-BF reduces the FPR by 81% compared to LBF or sandwiched LBF (disjoint FPR reduces the FPR by 84%). Moreover, to achieve a FPR $\approx 0.9\%$, Ada-BF and disjoint Ada-BF only require 200Kb, while both LBF and the sandwiched LBF needs more than 350Kb. And to get a FPR $\approx 0.35\%$, Ada-BF and disjoint Ada-BF reduce the memory usage from over 500Kb of LBF to 300Kb, which shows that our proposed algorithms save over 40% of the memory usage compared with LBF and sandwiched LBF.

---

[1]The Random Forest classifier consists 10 decision trees, and each tree has at most 20 leaf nodes.

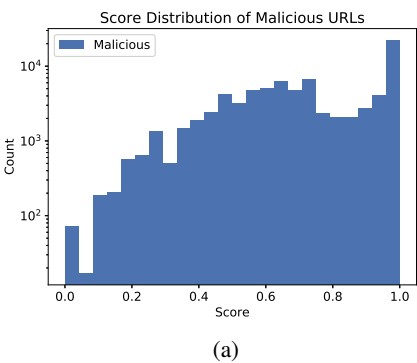
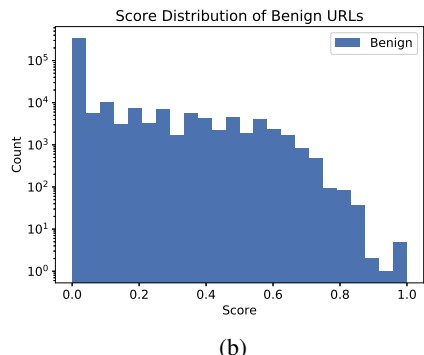

(a)  (b)

Figure 2: Histogram of the classifier's score distributions of keys (Malicious) and non-keys (Benign) for Task 1. We can see that $n_j$ (number of keys in region $j$) is monotonic when score > 0.3. The partition was only done to ensure $\frac{p_j}{p_j+1} \geq c$

## 5.2  TASK 2: VIRUS SCAN

Bloom filter is widely used to match the file's signature with the virus signature database. Our dataset includes the information of 41323 benign files and 96724 viral files. The virus files are collected from VirusShare database (Vir). The dataset provides the MD5 signature of the files, legitimate status and other 53 variables characterizing the file, like "Size of Code", "Major Link Version" and "Major Image Version". We trained a machine learning model with these variables to differentiate the benign files from the viral documents. We randomly selected 20% samples as the training set to build a binary classification model using Random Forest model [2]. We used "sklearn.ensemble.RandomForestClassifier" to tune the model, and the Random Forest classifier costs about 136Kb. The classification model achieves 0.98 prediction accuracy on the testing set. The predicted the class probability (with the function "predict_prob" in "sklearn" library) is used as the score $s(x)$. Other implementation details are similar to that in Task 1.

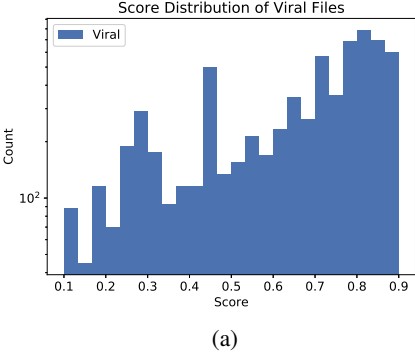
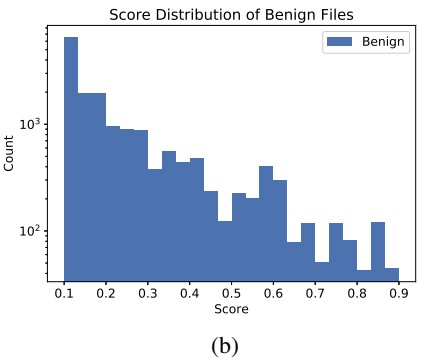

(a)  (b)

Figure 3: Histogram of the classifier score distributions for the Virus Scan Dataset. The partition was only done to ensure $\frac{p_j}{p_j+1} \geq c$.

**Result:**  As the machine learning model achieves high prediction accuracy, figure 4 suggests that all the learned Bloom filters show huge advantage over the standard BF where the FPR is reduced by over 98%. Similar to the previous experiment results, we observe consistently lower FPRs of our algorithms although the the score distributions are not smooth or continuous (Figure 3). Again, our methods show very similar performance. Compared with LBF, our methods reduce the FPRs

---

[2]The Random Forest classifier consists 15 decision trees, and each tree has at most 5 leaf nodes.

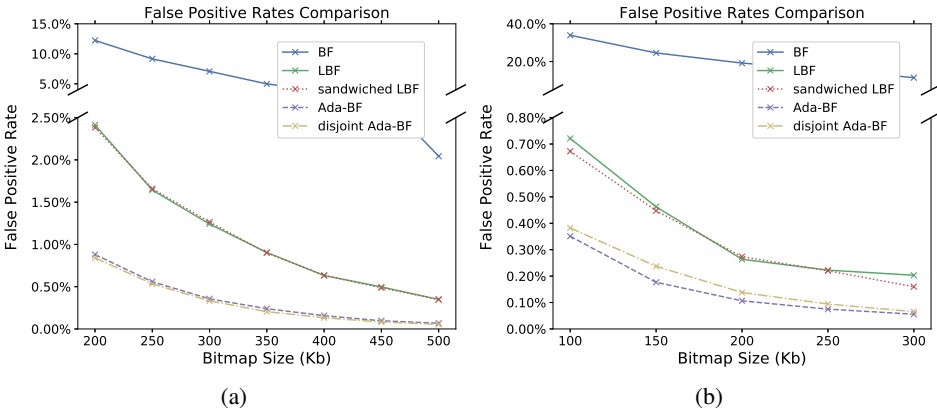

(a)                                                                              (b)

Figure 4: FPR with memory budget for all the five baselines (the bit budget of BF = bitmap size + learner size). (a) FPRs comparison of Malicious URL detection experiment; (b) FPRs comparison of Virus scan experiment.

by over 80%. To achieve a 0.2% FPR, the LBF and sandwiched LBF cost about 300Kb bits, while Ada-BF only needs 150Kb bits, which is equivalent to 50% memory usage reduction compared to the previous methods.

### 5.3    SENSITIVITY TO HYPER-PARAMETER TUNING

Compared with the LBF and sandwiched LBF where we only need to search the space of $\tau$ to optimize the FPR, our algorithms require to tune a series of score thresholds. In the previous sections, we have proposed a simple but useful tuning strategies where the score thresholds can be determined by only two hyper-parameters, $(K, c)$. Though our hyper-parameter tuning technique may lead to a sub-optimal choice, our experiment results have shown we can still gain significantly lower FPR compared with previous LBF. Moreover, if the number of groups $K$ is misspecified from the optimal choice (of $K$), we can still achieve very similar FPR compared with searching both $K$ and $c$. Figure 5 shows that for both Ada-BF and disjoint Ada-BF, tuning $c$ while fixing $K$ has already achieved similar FPRs compared with optimal case by tuning both $(K, c)$, which suggests our algorithm does not require very accurate hyper-parameter tuning to achieve significant reduction of the FPR.

### 5.4    DISCUSSION: SANDWICHED LEARNED BLOOM FILTER VERSUS LEARNED BLOOM FILTER

Sandwiched LBF is a generalization of LBF and performs no worse than LBF. Although Mitzenmacher (2018) has shown how to allocate bits for the initial filter and backup filter to optimize the expected FPR, their result is based on the a fixed FNR and FPR. While for many classifiers, FNR and FPR are expressed as functions of the prediction score $\tau$. Figure 4(a) shows that the sandwiched LBF always has the same FPR as LBF though we increase the bitmap size from 200Kb to 500Kb. This is because the sandwiched LBF is optimized when $\tau$ corresponds to a small FPR and a large FNR, where the optimal backup filter size even exceeds the total bitmap size. Hence, we should not allocate any bits to the initial filter, and the sandwiched LBF reduces to LBF. On the other hand, our second experiment suggests as the bitmap size becomes larger, sparing more bits to the initial filter is clever, and the sandwiched LBF shows the its advantage over the LBF (Figure 6(b)).

## 6    CONCLUSION

We have presented new approaches to implement learned Bloom filters. We demonstrate analytically and empirically that our approaches significantly reduce the FPR and save the memory usage compared with the previously proposed LBF and sandwiched LBF even when the learner's discrimination power . We envision that our work will help and motivate integrating machine learning model into probabilistic algorithms in a more efficient way.

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

## APPENDIX A  SENSITIVITY TO HYPER-PARAMETER TUNING

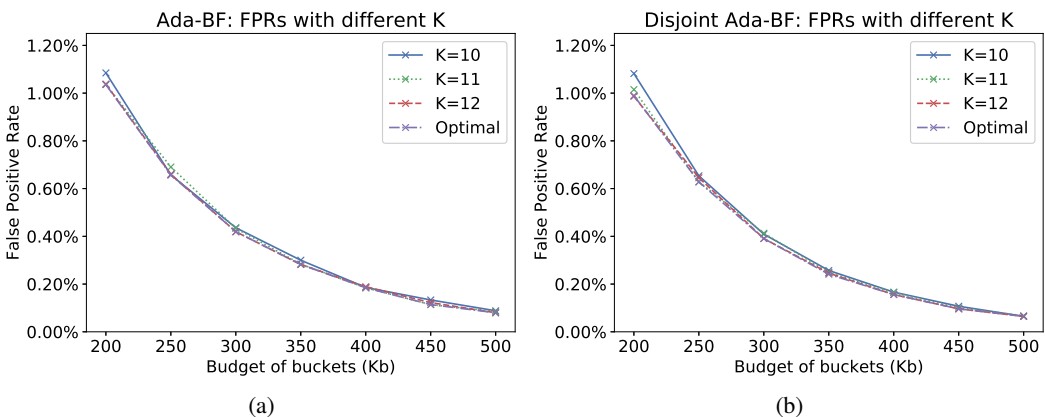

Figure 5: FPR comparison of tuning $c$ while fixing the number of groups $K$ and tuning both $(K, c)$

## APPENDIX B  MORE COMPARISONS BETWEEN THE LBF AND SANDWICHED LBF

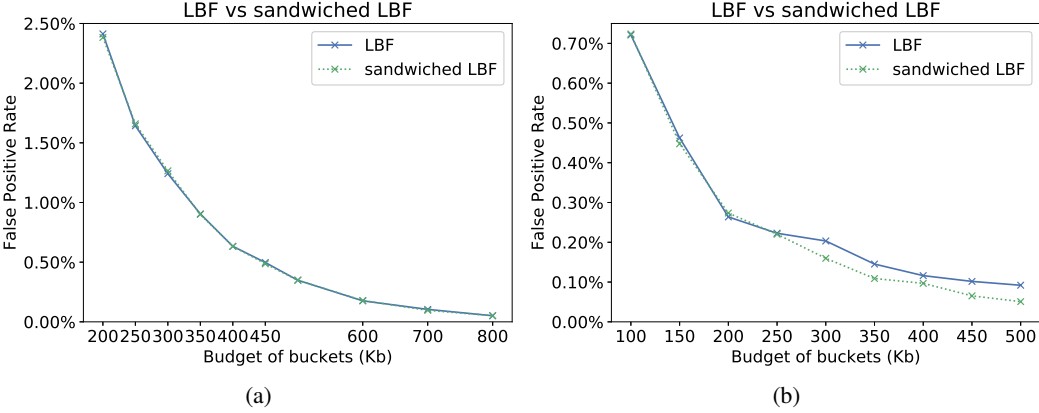

Figure 6: FPR comparison between LBF and sandwiched LBF under different bitmap sizes. (a) malicious URL experiment; (b) malware detection experiment

## APPENDIX C  COMPARING THE BLOOM FILTER TO HIERARCHICAL HASHING

The machine learning model used in the learned Bloom filters is critical because it has discrimination power between the keys and non-keys and is more efficient in identifying keys in some cases. To show its unique role, we replaced the machine learning model with another Bloom filter such that it becomes a hierarchical Bloom filter (learner is replaced by an initial filter). To implement the hierarchical Bloom filter, we spare 50% of the bit budget to the initial filter and use the other bits to build the backup filter.

Figure 7 shows that the hierarchical BF does not outperform the original BF under all the budget of buckets, and in some cases, it even achieves a worse FPR. Hence, using a random hash function to replace the learner is not a memory efficient approach.

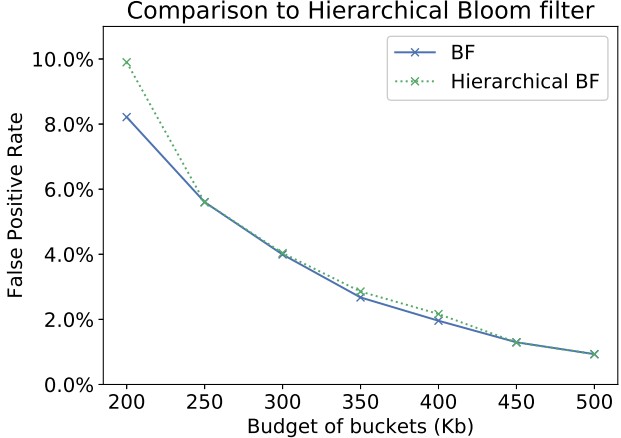

Figure 7: FPR comparison between LBF and sandwiched LBF under different bitmap sizes. (a) malicious URL experiment; (b) malware detection experiment

## APPENDIX D   PROOF OF THE STATEMENTS

**Proof of Lemma 1:** Let $Z_j(x) = \sum_{i=1}^{m} \mathbb{1}(s(x) \in [\tau_{j-1}, \tau_j) | x \notin S)$, then $Z_j(x) \sim Bernoulli(p_j)$, and $m_j = \sum_{i=1}^{m} Z_j(x_i)$ counts the number of non-keys falling in group $j$ and $\hat{p}_j = \frac{m_j}{m}$. To upper bound the probability of the overall estimation error of $p_j$, first, we need to evaluate its expectation, $\mathbb{E}\left(\sum_{j=1}^{K} |\hat{p}_j - p_j|\right)$.

Since $m_j$ is a binomial random variable, its exact cdf is hard to compute. But with central limit theorem, when $m$ is large, $\frac{m_j - mp_j}{\sqrt{mp_j(1-p_j)}} \longrightarrow N(0,1)$. Thus, we can approximate $\mathbb{E}\left(|\hat{p}_j - p_j|\right) = \mathbb{E}\left(|\frac{m_j - mp_j}{\sqrt{mp_j(1-p_j)}}|\right) \cdot \sqrt{\frac{p_j(1-p_j)}{m}} \approx \sqrt{\frac{2}{\pi}} \cdot \sqrt{\frac{p_j(1-p_j)}{m}}$ (if $Z \sim N(0,1)$, $\mathbb{E}(|Z|) = \sqrt{\frac{2}{\pi}}$). Then, the expectation of overall error is approximated by $\mathbb{E}\left(\sum_{j=1}^{K} |\hat{p}_j - p_j|\right) \approx \sqrt{\frac{2}{m\pi}} \cdot \left(\sum_{j=1}^{K} \sqrt{p_j(1-p_j)}\right)$, which goes to $0$ as $m$ becomes larger.

We need to further upper bound the tail probability of $\sum_{j=1}^{K} |\hat{p}_j - p_j|$. First, we upper bound the variance of $\sum_{j=1}^{K} |\hat{p}_j - p_j|$,

$$\mathrm{Var}\left(\sum_{j=1}^{K} |\hat{p}_j - p_j|\right) \leq K \sum_{j=1}^{K} \mathrm{Var}\left(|\hat{p}_j - p_j|\right) = K \sum_{j=1}^{K} \left(\mathrm{Var}\left(\hat{p}_j - p_j\right) - \mathbb{E}\left(|\hat{p}_j - p_j|\right)^2\right)$$

$$\approx \frac{K}{m} \sum_{j=1}^{K} \left(p_j(1-p_j) - \frac{2}{\pi}\left(\sum_{i=1}^{K} \sqrt{p_j(1-p_j)}\right)^2\right) \triangleq \frac{K}{m} V(\mathbf{p})$$

Now, by envoking the Chebyshev's inequality,

$$\mathrm{P}\left[\sum_{j=1}^{K} |\hat{p}_j - p_j| \geq \epsilon\right] = \mathrm{P}\left[\sum_{j=1}^{K} |\hat{p}_j - p_j| - \mathbb{E}\left(\sum_{j=1}^{K} |\hat{p}_j - p_j|\right) \geq \epsilon - \mathbb{E}\left(\sum_{j=1}^{K} |\hat{p}_j - p_j|\right)\right]$$

$$\leq \frac{\mathrm{Var}\left(\sum_{j=1}^{K} |\hat{p}_j - p_j|\right)}{\left(\epsilon - \mathbb{E}\left(\sum_{j=1}^{K} |\hat{p}_j - p_j|\right)\right)^2}$$

$$= \frac{KV(\mathbf{p})}{m\left(\epsilon - \mathbb{E}\left(\sum_{j=1}^{K} |\hat{p}_j - p_j|\right)\right)^2} \longrightarrow 0 \text{ as } m \longrightarrow \infty$$

Thus, $\sum_{j=1}^{K}|\hat{p}_j - p_j|$ converges to 0 in probability as $m \longrightarrow \infty$. $\square$

Moreover, since we have

$$\mathbb{E}\left(\sum_{j=1}^{K}|\hat{p}_j - p_j|\right) \approx \sqrt{\frac{2}{m\pi}}(\sum_{j=1}^{K}\sqrt{p_j(1-p_j)}) \leq \sqrt{\frac{2}{m\pi}(K-1)} \qquad (4)$$

$$V(\mathbf{p}) = \sum_{j=1}^{K}\left(p_j(1-p_j) - \frac{2}{\pi}\left(\sum_{i=1}^{K}\sqrt{p_j(1-p_j)}\right)^2\right)$$

$$\leq \sum_{j=1}^{K}\left(p_j(1-p_j)\left(1-\frac{2}{\pi}\right)\right)$$

$$\leq \left(1-\frac{2}{\pi}\right)\left(1-\frac{1}{K}\right) \qquad (5)$$

Then, by Eq 4 and Eq 5, we can upper bound $P\left[\sum_{j=1}^{K}|\hat{p}_j - p_j| \geq \epsilon\right]$ by,

$$P\left[\sum_{j=1}^{K}|\hat{p}_j - p_j| \geq \epsilon\right] \leq \frac{KV(\mathbf{p})}{m\left(\epsilon - \mathbb{E}\left(\sum_{j=1}^{K}|\hat{p}_j - p_j|\right)\right)^2}$$

$$\leq \frac{(1-\frac{2}{\pi})(K-1)}{m\left(\epsilon - \sqrt{\frac{2}{m\pi}(K-1)}\right)^2} \qquad (6)$$

When $m \geq \frac{2(k-1)}{\epsilon^2}\left[\sqrt{\frac{1}{\pi}} + \sqrt{\frac{1-2/\pi}{\delta}}\right]^2$, we have $m\left(\epsilon - \sqrt{\frac{2}{m\pi}(K-1)}\right)^2 \geq \frac{(K-1)(1-\frac{2}{\pi})}{\delta}$, thus, $P\left[\sum_{j=1}^{K}|\hat{p}_j - p_j| \geq \epsilon\right] \leq \delta$. $\square$

**Proof of Theorem 1:**  For comparison, we choose $\tau = \tau_{g-1}$, for both LBF and Ada-BF, queries with scores larger than $\tau$ are identified as keys directly by the same machine learning model. Thus, to compare the overall FPR, we only need to evaluate the FPR of queries with score lower than $\tau$.

Let $p_0 = P[s(x) < \tau|x \notin S]$ be the probability of a key with score lower than $\tau$. Let $n_0$ denote the number of keys with score less than $\tau$, $n_0 = \sum_{i:x_i \in S} I(s(x_i) < \tau)$. For learned Bloom filter using $K$ hash functions, the expected FPR follows,

$$\mathbb{E}(\text{FPR}) = (1-p_0) + p_0\left(1 - \left(1 - \frac{1}{R}\right)^{Kn_0}\right)^K = 1 - p_0 + p_0\beta^K, \qquad (7)$$

where $R$ is the length of the Bloom filter. For Ada-BF, assume we fix the number of groups $g$. Then, we only need to determine $K_{max}$ and $K_{min} = K_{max}-g+1$. Let $p_j = Pr(\tau_{j-1} \leq s(x) < \tau_j|x \notin S)$ The expected FPR of the Ada-BF is,

$$\mathbb{E}(\text{FPR}_a) = \sum_{j=1}^{g}p_j\left(1 - \left(1 - \frac{1}{R}\right)^{\sum_{j=1}^{g-1}K_j n_j}\right)^K = \sum_{j=1}^{g-1}p_j\alpha^{K_j}, \qquad (8)$$

where $\sum_{j=1}^{g-1}n_j = n_0$. Next, we give a strategy to select $K_{max}$ which ensures a lower FPR of Ada-BF than LBF.

Select $K_{max} = \lfloor K + \frac{g}{2} - 1 \rfloor$. Then, we have

$$
\begin{aligned}
n_0 K &= \sum_{j=1}^{g-1} n_j K = K \left[ n_1 + \sum_{i=2}^{g-1}(n_1 + \sum_{i=1}^{j-1} T_i) = n_1(g-1) + \sum_{j=1}^{g-2} T_j(g-j-1) \right] \\
&= \frac{2K}{g-2} \left[ \frac{(g-1)(g-2)}{2} n_1 + \sum_{j=1}^{g-2} \frac{(g-2)(g-1-j)}{2} T_j \right] \\
&\leq \frac{2}{g-2} \left[ \frac{(g-1)(g-2)}{2} n_1 + \sum_{j=1}^{g-2} \frac{(g+j-2)(g-1-j)}{2} T_j \right] \\
&= \frac{2}{g-2} \sum_{j=1}^{g-1}(j-1) n_j
\end{aligned}
\tag{9}
$$

By Eq 9. we further get the relationship between $\alpha$ and $\beta$.

$$
\sum_{j=1}^{g-1} K_j n_j = \sum_{j=1}^{g-1}(K_{max} - j + 1) n_j \leq n_0 \left( K_{max} - \frac{g}{2} + 1 \right) \leq n_0 K \implies \alpha \leq \beta.
$$

Moreover, by Eq. 3, we have,

$$
\begin{aligned}
\mathbb{E}(\text{FPR}_a) = \frac{(1-c)(1-(c\alpha)^g)}{(\frac{1}{\alpha}-c)(\alpha^g - (c\alpha)^g)} \alpha^{K_{max}} &\leq \frac{(1-c)(1-(c\alpha)^g)}{(\frac{1}{\alpha}-c)(\alpha^g - (c\alpha)^g)} \beta^{K_{max}} \\
&\leq \beta^{K_{max}} \frac{\alpha(c-1)}{c\alpha - 1} \\
&< \mathbb{E}(\text{FPR}) \left( \frac{1+\lambda}{\lambda} \beta^{K_{max}-K} \right) \\
&\leq \mathbb{E}(\text{FPR}) \left( \frac{1+\lambda}{\lambda} \beta^{\lfloor g/2-1 \rfloor} \right).
\end{aligned}
$$

Therefore, as $g$ increases, the upper bound of $\mathbb{E}(\text{FPR}_a)$ decreases exponentially fast. Moreover, since $\frac{1+\lambda}{\lambda}$ is a constant, when $g$ is large enough, we have $\frac{1+\lambda}{\lambda} \beta^{\lfloor g/2-1 \rfloor} \leq 1$. Thus, the $\mathbb{E}(\text{FPR}_e)$ is reduced to strictly lower than $\mathbb{E}(\text{FPR})$. $\square$

**Proof of Theorem 2:** Let $\eta = \frac{\log(c)}{\log(\mu)} \approx \frac{\log(c)}{\log(0.618)} < 0$. By the tuning strategy described in the previous section, we require the expected false positive items should be similar across the groups. Thus, we have

$$
p_1 \cdot \mu^{R_1/n_1} = p_j \cdot \mu^{R_j/n_j} \implies R_j = n_j \left( \frac{R_1}{n_1} + (j-1)\eta \right), \quad \text{for } j \in [g-1]
$$

where $R_j$ is the budget of buckets for group $j$. For group $j$, since all the queries are identified as keys by the machine learning model directly, thus, $R_g = 0$. Given length of Bloom filter for group 1, $R_1$, the total budget of buckets can be expressed as,

$$
\sum_{j=1}^{g-1} R_j = \sum_{j=1}^{g-1} \frac{n_j}{n_1} R_1 + (j-1) n_j \eta
$$

Let $p_0 = Pr(s(x) < \tau | x \notin S)$ and $p_j = Pr(\tau_{j-1} \leq s(x) < \tau_j | x \notin S)$. Let $n_0$ denote the number of keys with score less than $\tau$, $n_0 = \sum_{i:x_i \in S} I(s(x_i) < \tau)$, and $n_j$ be the number of keys in group

$j$, $n_j = \sum\limits_{i:x_i \in S} I(\tau_{j-1} \leq s(x_i) < \tau_j)$. Due to $\tau = \tau_{g-1}$, we have $\sum_{j=1}^{g-1} n_j = n_0$. Moreover, since $\tau_{g-1} = \tau$, queries with score higher than $\tau$ have the same FPR for both disjoint Ada-BF and LBF. So, we only need to compare the FPR of the two methods when the score is lower than $\tau$. If LBF and Ada-BF achieve the same optimal expected FPR, we have

$$
\begin{aligned}
p_0 \cdot \mu^{R/n_0} &= \sum_{j=1}^{g-1} p_j \cdot \mu^{R_j/n_j} = g \cdot p_1 \cdot \mu^{R_1/n_1} \\
\implies R &= \frac{n_0}{n_1} R_1 - n_0 \frac{\log(p_0/p_1) - \log(g)}{log(\mu)} \\
&= \sum_{j=1}^{g-1} \left[ \frac{n_j}{n_1} R_1 - n_j \frac{\log(1 - \left(\frac{1}{c}\right))^g - \log\left(1 - \frac{1}{c}\right) - \log(g)}{\log(\mu)} \right],
\end{aligned}
$$

where $R$ is the budget of buckets of LBF. Let $T_j = n_{j+1} - n_j \geq 0$. Next, we upper bound $\sum_{j=1}^{g-1} n_j$ with $\sum_{j=1}^{g-1} (j-1) n_j$.

$$
\begin{aligned}
\sum_{j=1}^{g-1} n_j &= n_1 + \sum_{i=2}^{g-1} \left( n_1 + \sum_{i=1}^{j-1} T_i \right) = n_1(g-1) + \sum_{j=1}^{g-2} T_j(g-j-1) \\
&= \frac{2}{g-2} \left[ \frac{(g-1)(g-2)}{2} n_1 + \sum_{j=1}^{g-2} \frac{(g-2)(g-1-j)}{2} T_j \right] \\
&\leq \frac{2}{g-2} \left[ \frac{(g-1)(g-2)}{2} n_1 + \sum_{j=1}^{g-2} \frac{(g+j-2)(g-1-j)}{2} T_j \right] \\
&= \frac{2}{g-2} \sum_{j=1}^{g-1} (j-1) n_j
\end{aligned}
$$

Therefore, we can lower bound $R$,

$$
R \geq \sum_{j=1}^{g-1} \left[ \frac{n_j}{n_1} R_1 - (j-1) n_j \frac{2(\log(1 - \left(\frac{1}{c}\right))^g - \log\left(1 - \frac{1}{c}\right) - \log(g))}{(g-2)\log(\mu)} \right].
$$

Now, we can lower bound $R - \sum_{j=1}^{g-1} R_j$,

$$
R - \sum_{j=1}^{g-1} R_j \geq \sum_{j=1}^{g-1} (j-1) n_j \left[ -\eta - \frac{2(\log(1 - \left(\frac{1}{c}\right))^g - \log\left(1 - \frac{1}{c}\right) - \log(g))}{(g-2)\log(\mu)} \right].
$$

Since $\eta$ is a negative constant, while $\frac{2(\log(1-\left(\frac{1}{c}\right))^g - \log\left(1-\frac{1}{c}\right) - \log(g))}{(g-2)\log(\mu)}$ approaches to 0 when $g$ is large. Therefore, when $g$ is large, $\eta - \frac{2(\log(1-\left(\frac{1}{c}\right))^g - \log\left(1-\frac{1}{c}\right) - \log(g))}{(g-2)\log(\mu)} < 0$ and $R - \sum_{j=1}^{g-1} R_j$ is strictly larger than 0. So, disjoint Ada-BF consumes less memory than LBF to achieve the same expected FPR.

