# OpenReview forum: "Adaptive Learned Bloom Filter (Ada-BF): Efficient Utilization of the Classifier"
_ICLR.cc/2020/Conference — Reject_

### Official Review · AnonReviewer2 · 2019-10-23
**Official Blind Review #2**

**Rating:** 3

**Review:**

The paper proposed an adaptive learned bloom filter. Rather than setting a threshold of prediction score, the paper partitions the score into several intervals; for query insider each interval, the paper either uses a group of independent hash functions to hash the query in one unified bloom filter or introduce an independent bloom filter. The paper proposes efficient ways to tune the hyper-parameters, and provides the analysis of the error. Experiments on two applications show the effectiveness of the proposed methods.

The idea seems useful. However, I have a few concerns/questions. My decision depends on the responses of the author(s).

(1) Although the paper claims the score information can be fully exploited, the paper seems to do hashing for all possible queries. Why not set up a tau such that only when the score is below tau, we conduct the adaptive hashing? When the score is bigger than tau, we still claim an identification of the key? In this way, a bunch of keys can still be saved and without extra hashing.

(2) The proposed method seems to be a hierarchical hashing strategy.  The first level is to hash the queries into different intervals through a score function learned from data. Why not compose another group of random hash to do the first level hashing? What is the major benefit of collecting training examples to run a machine learning model? Accordingly, why not compare with such a baseline using a group of random hashing to do the first level?

**Experience Assessment:**

I do not know much about this area.

**Review Assessment: Checking Correctness Of Derivations And Theory:**

I assessed the sensibility of the derivations and theory.

**Review Assessment: Checking Correctness Of Experiments:**

I assessed the sensibility of the experiments.

**Review Assessment: Thoroughness In Paper Reading:**

I read the paper at least twice and used my best judgement in assessing the paper.

---

> ### Author Response · Authors · 2019-11-12
> **Response to Reviewer 2**
>
> Thanks for giving us the chance to address your misunderstandings and being open to change in scores!
>
> The machine learning model is critical because it has discrimination power between the keys and non-keys, which cannot be replaced by a random hash function. Hierarchical random hashing is not any different from one random hashing, in fact, it is a little worse as hierarchy leads to biased randomness (For more details, please refer to the related literature on tabulation based hashing by Mikkel Thorup). So, hierarchical random hashing can be at best (assuming entropy in the data) as good as the vanilla Bloom filter. Since we have shown the vanilla Bloom filter is worse than Ada-BF in our paper, hierarchical random hashing does not outperform our algorithms.
>
> On the other hand, since the discrimination power of the machine learning model is independent of the size of queries, a learner can be more efficient for membership tests compared with the Bloom filter in many cases. For example, if thousands of malicious URLs include the pattern ‘abcabc’, then for a decision tree, it only takes several bits to remember this pattern while for the Bloom filter, it may take thousands of bits to ‘remember’ these thousands of URLs with the pattern ‘abcabc’ as its memory usage grows with the size of queries. Clearly, when the queries can be classified with simple patterns that are easy to be learned by a machine learning model, incorporating a learner to the Bloom filter saves more bits.
>
> Regarding the number of hash functions used in the high score group, we set the minimum number of hash functions to 0 for the group with the highest scores (Page 4, before lemma 1) for both of our experiments, where the membership testing purely relies on the machine learning model.  So, we were doing exactly the same thing as you mentioned in your first concern.
>
> Ada-BF generalizes the learned Bloom filter, and the number of hash functions in the backup filter varies across different groups.  Our hyper-parameter tuning strategy requires setting the minimum and the maximum number of hash functions. When the learner provides an accurate prediction for the queries falling in the high score region, we may just set the minimum number of hash functions to 0 to save the memory usage and reduce the overall false-positive rate.

---

> > ### Author Response · Authors · 2019-11-15
> > **Further Experiments on Hierarchical Hashing**
> >
> > We implemented hierarchical random hashing on the malicious URL dataset (for more details, please see the appendix C figure 7). We spared 50% of the bit budget to the initial Bloom filter and used the other bits to build the backup filter. It shows that the hierarchical Bloom filter does not outperform the standard Bloom filter under all the bit budgets. In some cases,  it even reaches a worse false positive rate. Hence, our experiment results validate the reasoning in the previous paragraphs, suggesting the critical role of the machine learning model in Ada-BF (and other learned Bloom filters).

---

### Official Review · AnonReviewer3 · 2019-10-23
**Official Blind Review #3**

**Rating:** 6

**Review:**

This paper extends the Bloom filter learning by using the complete spectrum of the scores regions. It uses multiple thresholds and then varies the number of hash functions among different scores regions to obtain better trade-off. Detailed theoretical analysis provides guaranteed superiority over learned Bloom filter under some conditions. The experiments also show the two proposed methods outperform learned Bloom filter in FPR and memory usage.

The motivation is based on the observation that the keys' score distribution density has an opposite trend to the non-keys. Though the experiment results support this observation, some theoretical analysis on this and its relationship with the final FPR could be provided.

**Experience Assessment:**

I do not know much about this area.

**Review Assessment: Checking Correctness Of Derivations And Theory:**

I assessed the sensibility of the derivations and theory.

**Review Assessment: Checking Correctness Of Experiments:**

I assessed the sensibility of the experiments.

**Review Assessment: Thoroughness In Paper Reading:**

I read the paper thoroughly.

---

> ### Author Response · Authors · 2019-11-12
> **Response to Reviewer 3**
>
> Thanks for your thoughtful comments and support for the paper!
>
> As you pointed out, the success of Ada-BF is based on the observation that the keys' score distribution density has an opposite trend compared to the non-keys. This observation is quite general since when the learner has a good discrimination power, the score distribution of keys and non-keys cannot be similar. Our algorithms make full use of this observation. Moreover, our experiment 2 also shows that even when the score density is a bit noisy, both Ada-BF and disjoint Ada-BF still outperform other state-of-the-art methods by a great margin, suggesting their robustness to the noisy score distributions.

---

### Official Review · AnonReviewer1 · 2019-10-24
**Official Blind Review #1**

**Rating:** 6

**Review:**

This paper proposes two new bloom filter algorithms that incorporate a learnt model for estimating if an input is in the set or not. These methods blend the space between pure BF and learnt BF with one threshold by creating regions over the score and having varying number of hash functions for each region.

I really like the paper and the approach taken. However, the experiments are on such small datasets that the true impact of these models aren't as impressive as they could be. In practice, BFs are used when dealing with millions/billions of entries to achieve real-time performance in real-world. For such applications, not only the memory is of concern but also the run-time part of the equation. In other words, if we have a learnt classifier for BF, how much will it impact the execution time vs memory usage as both are needed to be traded-off. It would have been great if the authors had experimented with larger datasets and had practical considerations for run-time and memory vs FPR investigated.

**Experience Assessment:**

I do not know much about this area.

**Review Assessment: Checking Correctness Of Derivations And Theory:**

I assessed the sensibility of the derivations and theory.

**Review Assessment: Checking Correctness Of Experiments:**

I carefully checked the experiments.

**Review Assessment: Thoroughness In Paper Reading:**

I read the paper at least twice and used my best judgement in assessing the paper.

---

> ### Author Response · Authors · 2019-11-12
> **Response to Reviewer 1**
>
> Thanks for your thoughtful comments and support for the paper!
>
> Our malicious URL experiment uses over half-a-million URLs which is a very reasonable size for  Bloom filters as used for caching malicious URLs in the web-browser.
>
> Regarding execution time and memory usage, weak classifiers (as used in the paper) can actually be more efficient than 2-universal hash functions, which is the case in our first experiment. We use 6 hash functions (bit array length = 500K) for vanilla Bloom filter. On a URL string, each of this hash function requires the operation of the order of the size of the string. Each of the 2-universal hash function is equivalent to inference (not training) with a linear classifier such as SVM or regression. In experiments, our classifier is a simple decision tree with less than 10 decisions based on simple features in the URL string. Clearly, the classifier operation is negligible and is even smaller than universal hashing if the object is heavy (like URL string).
>
> In our malicious URL experiment, the hashing operation (sklearn.utils.murmurhash3_32 in python) takes over 9s while the classifier inference takes less than 1s. So, our experiment shows the inference time is much shorter than the hashing time.
>
> It is the case that training the decision trees are slow but that is a one-time operation and hence is not a concern while querying.  So machine learning presents an interesting amortization tradeoff with a very slow one-time training process (like slower than other data structures) but super-efficient query, even faster than universal hash functions.

---

### Decision · Program_Chairs · 2019-12-19

**Decision:**

Reject

**Comment:**

The paper improves the Bloom filter learning by utilizing the complete spectrum of the scores regions.

The paper is nicely written with strong motivation and theoretical analysis of the proposed model. The evaluation could be improved: all the experiments are only tested on the small datasets, which makes it hard to assess the practicality of the proposed method. The paper could lead to a strong publication in the future if the issue on evaluation can be addressed.